# An Activation Likelihood Estimation Meta-Analysis of Voxel-Based Morphometry Studies of Chemotherapy-Related Brain Volume Changes in Breast Cancer

**DOI:** 10.3390/cancers17101684

**Published:** 2025-05-16

**Authors:** Sonya Utecht, Horacio Gomez-Acevedo, Jonathan Bona, Ellen van der Plas, Fred Prior, Linda J. Larson-Prior

**Affiliations:** 1Department of Biomedical Informatics, University of Arkansas for Medical Sciences, Little Rock, AR 72205, USA; gomezacevedohoracio@uams.edu (H.G.-A.); jpbona@uams.edu (J.B.); fwprior@uams.edu (F.P.); ljlarsonprior@uams.edu (L.J.L.-P.); 2Department of Pediatrics, University of Arkansas for Medical Sciences, Little Rock, AR 72205, USA; evanderplas@uams.edu; 3Department of Neuroscience, University of Arkansas for Medical Sciences, Little Rock, AR 72205, USA

**Keywords:** chemotherapy, chemotherapy-related cognitive impairment, cancer-related cognitive impairment, breast cancer, neuroimaging, magnetic resonance imaging (MRI), meta-analysis, voxel-based morphometry (VBM), brain volume, activation likelihood estimate (ALE)

## Abstract

Breast cancer patients and survivors exposed to chemotherapy face challenges with cognitive impairment, for which the underlying brain changes are not fully understood. Neuroimaging studies are exploring the structural and functional impacts of chemotherapy on the human brain but are limited by small sample sizes. Our study aims to synthesize volumetric neuroimaging data to highlight consistent findings in regional brain volume changes in breast cancer patients and survivors treated with chemotherapy.

## 1. Introduction

Chemotherapy has been shown to be associated with temporary and long-term cognitive impairment in many studies over the last several decades, with both self-reported cognitive issues and poorer performance compared to controls in multiple tests of executive function, attention, and processing speed [1,2,3,4,5,6,7]. However, the exact changes in structure and function of the brain that cause these changes are not fully understood. Additionally, there is not enough information on the specific chemotherapy regimens most associated with reported cognitive impairment nor the genetic and/or environmental factors that increase an individual’s chance of being affected. It is critical that we gather more information on these matters, as the use of chemotherapy has increased cancer survival rates, leaving many survivors for whom late effects of chemotherapy could impact quality of life.

Chemotherapy has been standard practice for invasive breast cancer for several decades, and the potential for neurocognitive damage has been under investigation for over forty years [8]. Many studies have shown that chemotherapy-exposed breast cancer survivors report more depression, anxiety, fatigue, and memory decline than age-matched control participants without cancer or breast cancer survivors who had not been exposed to chemotherapy [9]. Studies have also reported decreases in executive function and processing speed test scores based on neuropsychological tests such as the Mini-Mental State Exam, the Digit Span test, and the Stroop word test [10,11]. These tests are designed to assess general and specific mental functioning, such as memory, executive function, and processing speed. Subjects are also typically given questionnaires to determine self-perceived changes in cognitive function, memory, and mood [12,13,14]. Examples of these include the Behavioral Rating Inventory of Executive Function (BRIEF), the Clinical Assessment of Depression (CAD), and the Functional Assessment of Cancer Therapy-Cognitive Function (FACT-Cog) [15,16,17].

While cognitive impairments occur in a large number of patients [11,18,19], the mechanisms behind these impairments remain unclear. Oxidative stress, disruptions to the blood-brain barrier, genetic polymorphisms, and inflammation are all being investigated as possible mechanisms [20,21,22,23]. To better understand the causes of these neurological symptoms, researchers should examine quantitative changes in the brain. Therefore, neuroimaging provides a critical tool in the study of cancer-related cognitive impairment, as it provides opportunities to examine the structural and functional changes happening within the brain associated with exposure to chemotherapy. Changes in global or regional volume, surface thickness decreases, white matter integrity, blood flow changes, and changes in functional activation patterns can all be examined in minimally invasive ways with the use of different neuroimaging techniques and modalities.

However, while neuroimaging is a powerful tool for examining brain structure and function, many neuroimaging studies have sample sizes with between 9 and 60 subjects [24,25]. Small sample sizes in such studies make it difficult to gather enough statistical power to test hypotheses about chemotherapy effects.

Therefore, we performed a meta-analysis to synthesize findings from many brain structural studies and elucidate common discoveries of brain changes due to chemotherapy in breast cancer patients. Best practices for neuroimaging meta-analysis required that we narrow our focus to volumetric findings for consistency and comparability [26]. While this requirement makes pooling heterogeneous studies difficult and removes the option of using activation studies [26], it ensures that the resulting analysis fairly describes the findings in the current literature [27]. Brain regional volumes can be measured by segmenting regions of interest (ROIs) and measuring the total voxels or pixel thicknesses of the segmented area. However, differences in the chosen atlas can be inconsistent across segmented region-based studies. Another approach, voxel-based morphometry (VBM), has similar accuracy but does not require segmentation [28]. We therefore chose to perform an Activation Likelihood Estimation (ALE) meta-analysis [28,29,30,31,32] on VBM studies that matched our review criteria and provided coordinate data.

In a volume-focused VBM study, the brain is divided into 3D cubes mapped to a brain template [33]. The ‘MNI’ brain templates, originating from and named for the Montreal Neurological Institute, are prominent in the literature [34]. Voxels are assigned a value representing the density of the tissue in that location, modified by the values of the neighboring voxels for smoothing [33]. Comparison studies then examine and report the coordinates of the voxels of peak difference.

ALE techniques combine coordinates reported in multiple studies and report the regions of convergence [29,30]. The algorithm takes each coordinate in a single study, then treats it as the center point of a probability distribution, creating a modeled activation map [30]. The union of modeled activation maps from all of the included studies and the resulting coordinates are then tested against the null hypothesis that foci would be uniformly distributed across the brain [30]. The result is a spatial mapping of those areas the included studies agree show significant likelihood to be relevant. In our case, these will be the areas consistently found across studies to have decreased volume in chemotherapy-exposed breast cancer patients. The target question for our meta-analysis was “What areas of the brain consistently show volume reductions across studies in breast cancer patients and survivors exposed to chemotherapy?”. We believe the aforementioned methodology will allow us to better understand the brain changes underlying chemotherapy-related cognitive impairment.

## 2. Materials and Methods

Data gathering for this effort began in April of 2023 and was completed in November of 2023. We searched the following database repositories: Google Scholar, Embase, and Web of Science. Medline, the repository behind PubMed, also reports data to Google Scholar and Embase, so, while PubMed was not queried directly, manuscripts included there would be included in our search. Google Scholar also returns a broad range of results, sorting the more relevant results to the front; as such, only the first 100 results were examined in these queries due to decreasing relevance to the posed query. Eight distinct queries were used throughout the search process, narrowing the terms over time as the focus of the review took shape. While this change in search queries is not best practice, it became essential as fewer studies had comparable datasets than expected. Search query details can be found in Table 1.

The resulting manuscripts were then categorized and accepted or dismissed for the meta-analysis based on the following inclusion/exclusion criteria:

Inclusion:Journal article of a comparative neuroimaging study.Focus on cognitive impairment associated with breast cancer chemotherapy.Population consists of human female breast cancer patients and/or survivors that have started or completed one or more chemotherapy regimen(s).Comparison of population to themselves over time, other chemotherapy groups, or a control group.Volume changes investigated with voxel-based morphometry.Data provided in MNI or Talairach coordinate systems.

Exclusion:Reviews, abstracts, case studies, theses, books.Animal studies, cell line studies, fetal studies, or male cancer patients/survivors.Comparisons of treatments or mitigationsMachine learning classifiers.

The search queries provided a starting pool of 457 literature items after the removal of duplicates (Figure 1). Papers were then filtered based on their alignment with the criteria. Reviews, abstracts, and case reports were removed, followed by manuscripts that did not focus on breast cancer, chemotherapy, or neuroimaging. Treatment-focused manuscripts, AI classifier algorithm comparisons, and non-human studies were also removed. Finally, manuscripts were further filtered on the inclusion of comparable volumetric coordinate data.

Coordinate data were entered into a small custom database, stratified into experiments. In this SQLite database, each experiment was stored with a name, the number of subjects, the style of comparison, and the methodology. The database structure allowed for experiments to be queried by their methodology and comparison type to create input files specific to each comparison. The database structure can be viewed in the Appendix A. The largest group of data using the same methodology were the voxel-based morphometry (VBM) studies. There were 10 total VBM studies; however, one of these studies was removed because it included data pooled from other included studies. The remaining 9 studies include 11 experiments, which could then be divided into 2 groups: one comparing volumetric changes in the brains of breast cancer chemotherapy-positive patients (BCC+) over time (8 experiments) and the other comparing the peak differences in brain volume between BCC+ versus either healthy controls or breast cancer chemotherapy-negative patients (BCC−) (3 experiments) [10,35,36,37,38,39,40,41,42] We performed our meta-analysis with the GingerALE 3.0.2 software application by BrainMap (downloaded from brainmap.org on 16 May 2019) [29,30,31,43]. This application creates the unioned modeled activation maps from the input coordinates based on the ALE algorithms developed by Turkletaub and colleagues [28,29,30,32]. Multiple versions of the algorithms are available within the software; we used the 2009 Eickhoff algorithm [29]. A small Python program was developed to produce text files formatted for import into the ALE algorithm-performing software from our coordinate database. Two import files were created from our dataset, after which experiments using a single subject group were combined. One file consisted of the BCC+ over time and contained 26 coordinates, 232 subjects, and 8 experiments (Table 2).

The second file consisted of the BCC+ compared to controls, both breast cancer positive chemotherapy negative patients (BCC−) and non-cancer subjects (HC), and contains 20 coordinates and 251 subjects across 3 experiments (Table 3).

Each text file we created of coordinate data and associated subject count was loaded into the GingerALE application. ALE techniques take coordinate data and use them as probability distributions to create a map of the union of each voxel’s probability of activation, which is then checked for significance by comparison to a null distribution [29,30,43]. The null distribution that is compared against is configured by the GingerALE program using user-specified parameters. Our parameters were cluster-level FWE 0.0, threshold permutations: 1000, and *p* value: 0.001 as our configuration for both analyses. One coordinate in the BCC+ over time was out of the mask area and was therefore ignored by the algorithm. Configuration settings and coordinate text files can be found in the Appendix A.

## 3. Results

Our meta-analysis results showed that the right insula near the operculum (Figure 2A) and the left inferior frontal gyrus (Figure 2B) were most consistently found to have reduced volume over time among the breast cancer chemotherapy-positive patient groups in the selected voxel-based morphometry (VBM) studies. Other regions showing consistency across the breast cancer chemotherapy groups over time include the left medial frontal gyrus, superior frontal gyrus, anterior cingulate, right parahippocampal gyrus, right thalamus, left inferior parietal lobule, right superior parietal lobule, both superior temporal gyri, and the cerebellum (Figure 3). For each graphic, we have set the displayed cluster threshold to the value that most clearly shows the center points of the distributions.

We also ran an ALE meta-analysis on the BCC+ compared to the control data. Consistency was found in regions across the brain, including the left superior frontal gyrus, the right anterior cingulate, the right medial frontal gyrus, the right fusiform gyrus, and the right parahippocampus. However, the number of studies for this comparison was too small to show significance (Figure 4).

## 4. Discussion

Our meta-analysis, despite the small sample size, indicates that there is a high probability of volumetric changes in the bilateral insula and the left inferior frontal gyrus. While of lower probability, several other areas had convergence, such as the left medial frontal gyrus, right superior frontal gyrus, superior temporal gyri, anterior cingulate, right parahippocampal gyrus, cerebellum, left inferior parietal lobule, right superior parietal lobule, and the right thalamus post chemotherapy.

The brain is believed to operate as a set of functional networks that broadly encompass multiple brain regions. In evaluating the impact of reduced gray matter volume in specific brain regions, it is important to recognize that their relevance to changes in cognitive function is mediated by effects on the efficiency and function of brain networks. Relevant to changes in cognitive function, the frontoparietal, cingulo-opercular, salience, and default mode networks have garnered the most attention, as they are involved in higher-order cognitive function.

Our findings show that the insula was consistently reported to have reductions in volume in breast cancer patients who underwent chemotherapy. The insula is a core region of the cingulo-opercular network (CON), an executive network involved in task control that has been reported to show bilateral decreases in activation during verbal recall tasks [46], while another study reported decreased activation in the inferior frontal gyrus during encoding of a visual working memory task with increased activation in bilateral insula upon word recognition in BCC+ patients compared to controls [47]. Within-network connectivity between the bilateral insula and anterior cingulate was also reduced in BCC+ patients compared to healthy controls [47]. Other breast cancer cognitive impairment studies have found the insula to be correlated with inflammation and fatigue and lower memory scores on neurophysiological testing [38,48,49,50]. Recent studies have pointed to the complexity of the CON network [51,52] and to challenges in differentiating it from networks with similar anatomical signatures, such as the salience network.

The left inferior frontal gyrus, our other peak consistently reduced area, has been reported to show decreased connectivity to the hippocampus and reduced within-network connectivity in the frontoparietal network (FPN) within a week of chemotherapy [1]. The FPN is associated with executive functioning involving working memory, decision-making, and goal-oriented behavior. In another study, Piccirillo and colleagues found weakened inter-network connectivity between frontoparietal regions and cingulo-opercular networks in those reporting worsened cognitive function [53], indicating a loss of integration between executive control networks in breast cancer survivors exposed to chemotherapy.

We found consistent changes in the dorsolateral prefrontal cortex, precuneus, parahippocampus, anterior cingulate, and inferior parietal lobe. These areas are involved in the functional brain network known as the default mode network (DMN), which is involved in self-referential processing, future planning, and episodic memory. Multiple studies of connectivity have found alterations to this network in breast cancer chemotherapy survivors and patients. Dumas found that DMN within-network connectivity was reduced from baseline at the one-month post-chemotherapy mark and continued to be reduced at the one-year mark [24]. A later study by Feng found reduced connectivity between the anterior DMN and the CON [1]. In a study of older BCC+ women, Chen and colleagues found a decrease in resting-state blood oxygen dependent signal (SD-BOLD) variability in the posterior brain regions of chemotherapy survivors compared to age-matched controls, which correlated with worse composite scores on the National Institute of Health Toolbox Cognition Battery [54], particularly on the picture vocabulary tests, which supports their previous study finding alterations in the connectivity of the precuneus and DMN [55,56].

Consistent reductions in VBM studies were reported for areas in the medial temporal and superior parietal lobes, regions associated with the dorsal attention network (DAN). Multiple studies reported decreased connectivity in the DAN of BCC+ patients [25,57,58]. In a comparison of BCC+ patients based on self-reported cognitive issues, Kardan and collaborators found the between-network connectivity of the parietal and frontal regions was reduced following chemotherapy but exhibited recovery seven months post-chemotherapy [13]. Using diffusion tensor imaging (DTI) to study the integrity of white matter tracts, Deprez and colleagues found reduced fractional anisotropy (FA) across the brain, with parietal regions showing correlations with longer times on a test of processing and psychomotor speed [59]. Another study found that several regions involved in the attention networks had increased cerebral blood flow associated with lower scores during testing, which the authors suspect could be an unsuccessful and potentially harmful compensatory mechanism [60].

The brain networks impacted by the volumetric changes reported in this meta-analysis functionally track with the most common cognitive changes reported in breast cancer survivors who underwent chemotherapy, which largely center on higher-order processes such as memory and attention. Thus, as reported in both structural and functional studies, brain changes reported following chemotherapy in some breast cancer survivors are consistent with changes in cognitive function that can last well beyond anticipated recovery periods. To better understand those individuals at greater risk for chemotherapy-related cognitive decline, more consistency in study design is needed.

Recruiting enough patients into the multi-year longitudinal studies as required to gain the statistical power needed will continue to be difficult for single studies to achieve. Replication studies are generally less favored than studies using novel methodologies; however, in this field it will be necessary to verify findings from underpowered studies. To help mitigate these issues, data pooling has been suggested [61]; however, as with meta-analyses, this strategy is reliant on a large number of similar studies to aggregate. Consistency and replication must be improved to enable discovery of specific contributors to cognitive impairment in cancer survivors that can lead to actionable changes in patient treatment or post-chemotherapy interventions. The meta-analysis approach also runs the risk of publication bias [62]. An alternative solution would be to create multi-institutional consortia in which study protocols could be fully harmonized across study populations. It is critical that researchers within the field produce more consistent studies to improve analysis and synthesis. Replicating existing studies with new cohorts would also be a useful option to improve the ability for meta-analyses to be performed. In addition, many of the studies found in our review were coming from the same small number of labs, pointing to a need for better engagement of patients and patient-advocate groups to increase the number and diversity of interested patients across different regions and encourage growth in the number of researchers in this field [56]. In addition, future studies examining changes in brain regional volumes in BCC+ individuals should focus on more precise subnetwork identification to enable a clearer interpretation of the functional relevance of volume changes.

## 5. Conclusions

Chemotherapy is associated with both temporary and long-term cognitive impairments. To better understand these treatment side effects, we have examined physical brain changes that are visible with neuroimaging techniques. Because neuroimaging studies can be small, we chose to use a meta-analysis technique to bring data together from multiple studies, but even in so doing found that there was a limited amount of data. Nonetheless, the data currently available are sufficient to see large-scale brain effects in response to breast cancer chemotherapy, with primary changes reported in frontal gyri and the bilateral insula, which may underlie reductions in memory indexed by verbal recall tasks and lower memory scores on neurophysiological testing [38,48,49,50].

Improving consistency between studies to allow for more powerful meta-analysis, pooling data, replicating existing studies, and improving patient engagement will allow better exploration of variables among patients and survivors, such as chemotherapy agents, age, and genetic differences. Further research following these recommendations will produce better data, better results, and ultimately better outcomes for those facing cognitive issues from chemotherapy treatment in breast and other non-central nervous system cancers.

## Figures and Tables

**Figure 1 cancers-17-01684-f001:**
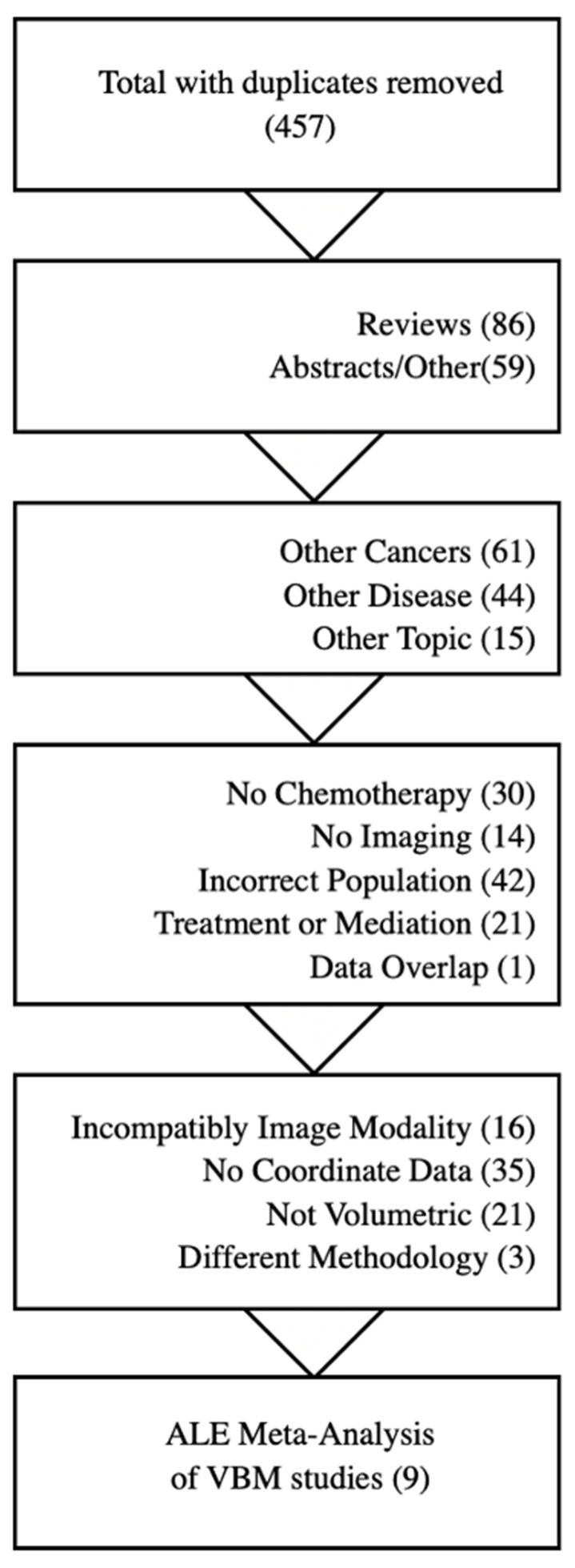
Literature search filtering for ALE meta-analysis.

**Figure 2 cancers-17-01684-f002:**
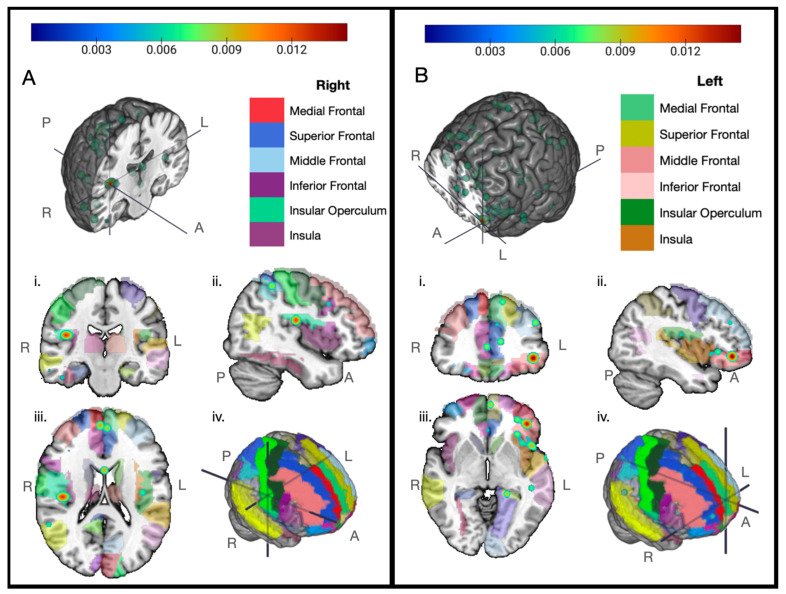
The two peak areas found most consistently correlated with changes in brain volume over time for breast cancer chemotherapy-positive subjects among eight included studies. Two regions show consistent peaks of volume change and are illustrated as a heat map from blue to red over a color legend indicating anatomical regions (automated anatomical labelling atlas [44]) on the Colin27 brain. Cluster threshold set to 0.0052749001 with red as the center peak for readability. Planes of section: i: Axial, ii: Sagittal, iii: Coronal, iv: 3D brain with peak section at the axis meeting point. A 3D model with the axial plane shown above. Image created with MRIcroGL [45]. Anatomical legend divided by hemisphere. R: right, L: left, A: anterior, P: posterior, (**A**) right insula, (**B**) left inferior frontal gyrus.

**Figure 3 cancers-17-01684-f003:**
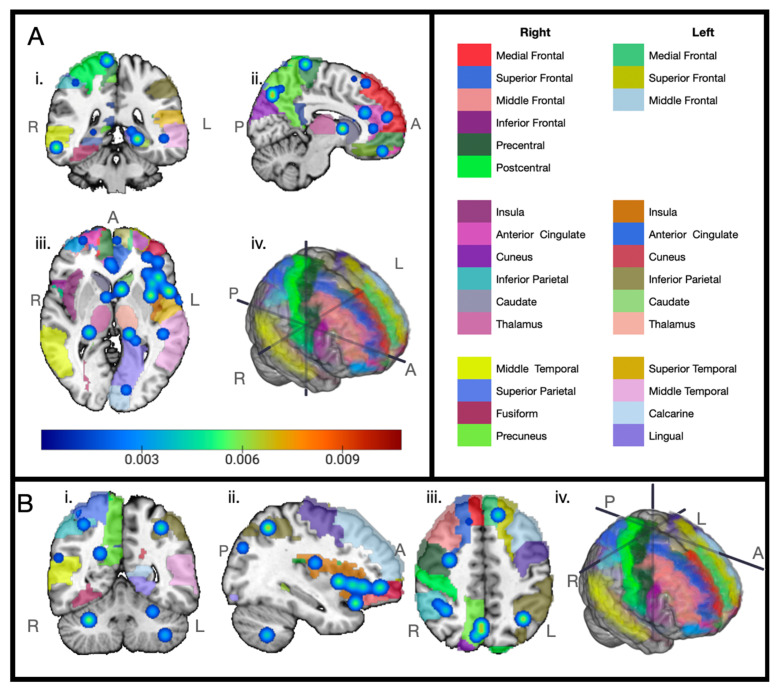
Areas consistently correlated with changes in brain volume over time for breast cancer chemotherapy-positive subjects between eight VBM studies. Regions of consistent findings illustrated as a heat map from blue to red over a color legend indicating anatomical regions (automated anatomical labeling atlas [44]) on the Colin27 brain. Cluster threshold set to 0.0012749001 for readability. Planes of section: i: axial, ii: sagittal, iii: coronal, iv: 3D brain showing meeting of plane sections. Image created with MRIcroGL [45]. Anatomical legend divided by hemisphere. R: right, L: left, A: anterior, P: posterior. (**A**) i: axial plane—right inferior temporal gyrus, right parietal inferior gyrus, right postcentral gyrus, right hippocampus, right parahippocampus, right medial temporal gyrus; ii: sagittal (right hemisphere)—right precuneus, right paracentral lobule, right medial frontal gyrus, right anterior cingulate, right caudate; iii: coronal plane—right middle frontal gyrus, right anterior cingulate, right caudate, right hippocampus, right thalamus, left medial frontal gyrus, left middle frontal gyrus, left inferior frontal gyrus, left superior temporal gyrus, left medial temporal gyrus, left insula, left caudate, left thalamus, left hippocampus, left parahippocampus, left lingual gyrus; iv: 3D brain showing meeting of plane sections. (**B**) i: axial plane—right middle temporal gyrus, right middle frontal gyrus, right inferior parietal gyrus, right precuneus, left inferior parietal gyrus, cerebellum; ii: sagittal plane (left hemisphere)—left inferior parietal gyrus, left insula, left inferior frontal gyrus, left middle frontal gyrus; iii: coronal plane—right inferior parietal gyrus, right precentral gyrus, superior medial frontal medial gyrus, right precuneus, left superior frontal gyrus, left inferior parietal gyrus; iv: 3D brain showing meeting of plane sections.

**Figure 4 cancers-17-01684-f004:**
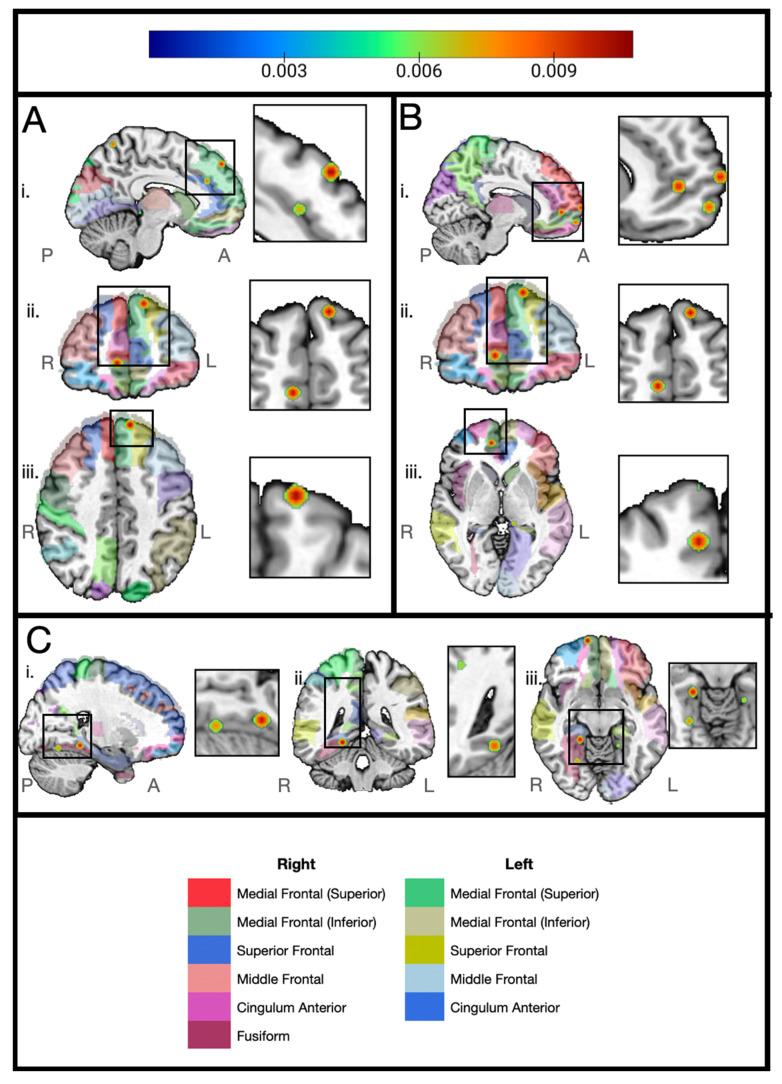
Areas in which breast cancer chemotherapy-positive subjects exhibit reduced volume compared to controls across three studies. Regions of consistent findings illustrated as a heat map from blue to red over a color legend indicating anatomical regions (automated anatomical labelling atlas [44]) on the Colin27 brain. Cluster threshold set to 0.0053741001 with red as the center peak for readability. Planes of section: i: sagittal, ii: axial, iii: coronal, iv: 3D brain showing meeting of plane sections. Image created with MRIcroGL [45]. Anatomical legend divided by hemisphere. R: right, L: left, A: anterior, P: posterior. (**A**) i: sagittal plane (left hemisphere)—left medial frontal gyrus; ii. axial plane—left medial frontal gyrus, right anterior cingulate; iii. coronal plane—left medial frontal gyrus, left superior frontal gyrus. (**B**) i: sagittal plane (right hemisphere)—right anterior cingulate, right superior frontal gyrus, right medial frontal gyrus; ii. axial plane—left medial frontal gyrus, right anterior cingulate; iii. coronal plane—right medial frontal. (**C**) i: sagittal plane (right hemisphere)—right parahippocampus, right fusiform gyrus; ii. axial plane—right parahippocampus; iii. coronal plane—bilateral parahippocampus, right fusiform gyrus.

**Table 1 cancers-17-01684-t001:** Search queries.

Repository	Terms	Date Performed	Filters
Google Scholar	“functional connectivity” and “chemotherapy” and “breast”	6 April 2023	First 100 results, Last 10 years
Google Scholar	“breast cancer” “cognitive impairment” “coordinates”	27 July 2023	First 100 results
Google Scholar	“neuroimaging” “breast cancer” “cognitive impairment”	27 July 2023	First 100 results
Google Scholar	(“matter volume” “breast cancer” “chemotherapy”)	3 September 2023	First 100 results
Web of Science	(‘chemotherapy) AND (‘breast cancer’) AND (‘neuroimaging’) AND (‘cognitive decline’ OR ‘cognitive decline’ OR ‘cognitive dysfunction’ OR ‘cognitive impairment’) AND (“gr$y matter” OR ‘white matter OR ‘brain volume’ or ‘volume’ or ‘volumetric’)	6 October 2023	
Embase	‘vbm’ AND (‘breast cancer’/exp OR ‘breast cancer’) AND (‘chemotherapy’/exp OR ‘chemotherapy’)	13 October 2023	
Embase	(‘breast cancer’/exp OR ‘breast cancer’) AND (‘chemotherapy’/exp OR ‘chemotherapy’) AND (‘neuroimaging’/exp OR ‘neuroimaging’) AND ‘cognitive defect’ AND [embase]/lim NOT ([embase]/lim AND [medline]/lim)	13 October 2023	Embase only
Embase	(‘breast cancer’/exp OR ‘breast cancer’) AND (‘brain volume’/exp OR ‘brain volume’)	1 November 2023	

**Table 2 cancers-17-01684-t002:** BCC+ studies selected for inclusion.

Authors	BCC+	Mean Age Mean (SD)	Coordinate Count	Chemotherapy Agent
McDonald 2010 [35]	17	52.4 (8.5)	24	AC-paclitaxel 12 TAC 2 AC 3 *
McDonald 2013 [36]	27	49.9 (7.6)	2	AC-T 9 TC 9 Tb 5 TAC 1 Paclitaxel 1 Not available 1 **
Lepage 2014 [37]	19	50.2 (8.6)	14	FEC-T 13 * TC 4 AC 1 AC-paclitaxel 1
Jenkins 2016 [38]	8	52.6 (3.9)	3	AC 1 FEC 2 FEC-T
Chen 2018 [39]	16	67(5.39)	9	TC 7 TbHP 1 paclitaxel/ trastuzumab 4 Carboplatin/paclitaxel 1 ddAC-paclitaxel 1 TAC 1
Li 2018 [40]	28	49.21 (8.15)	5	Doxorubicin and paclitaxel
Blommaert 2019 [10]	72	Younger group 43.7 (5.7) Older group 63.8 (3.4)	21	FEC 16 FEC-T 39
Zhou 2022 [41]	45	50.45 (9.08)	13	AC-T 11 EC-T 1 TAC 15 AT 1 TC 1 TbHP 11 TbHB 5

BCC+: Breast cancer chemotherapy-positive patients; SD: standard deviation; AC: doxorubicin/cyclophosphamide; AC-T: doxorubicin/cyclophosphamide followed by docetaxel; TAC: docetaxel/doxorubicin/cyclophosphamide; TC: docetaxel/cyclophosphamide; FEC: fluorouracil/epirubicin/cyclophosphamide; FEC-T: fluorouracil/epirubicin/cyclophosphamide followed by docetaxel; TC: Docetaxel/cyclophosphamide; Tb: Docetaxel/carboplatin; TbPH: Docetaxel/carboplatin/trastuzumab/pertuzmab; ddAC: dose-dense doxorubicin and cyclophosphamide; EC-T: epirubicin/cyclophosphamide followed by docetaxel; TbHB: docetaxel/carboplatin/trastuzumab/pyronib; * one patient also treated with trastuzumab; ** nine patients also treated with trastuzumab; one patient treated with bevacizumab.

**Table 3 cancers-17-01684-t003:** BCC+ vs. control (BCC− and HC) studies selected for inclusion.

Authors	Subjects	Coordinate Count	Chemotherapy Agent
Inagaki 2007 [42]	169	9	AC 3 CMF: 40 EC 2 Paclitaxel 2 tegafur/uracil 5
McDonald 2013 [36]	51	1	AC-T 9 TC 9 Tb 5 TAC 1 Paclitaxel 1 Not available 1 *
Chen 2018 [39]	31	9	TC 7 TbHP 1 paclitaxel/ trastuzumab 4 Carboplatin/paclitaxel 1 ddAC-paclitaxel 1 TAC 1

BCC+: Breast cancer chemotherapy-positive patients; BCC−: breast cancer patients that have not received chemotherapy; HC: Controls without breast cancer; AC: doxorubicin/cyclophosphamide; CMF: cyclophosphamide/methotrexate/fluorouracil; EC: epirubicin/cyclophosphamide; AC-T: doxorubicin/cyclophosphamide followed by docetaxel; TAC: docetaxel/doxorubicin/cyclophosphamide. TC: docetaxel/ cyclophosphamide; TC: Docetaxel/cyclophosphamide; Tb: Docetaxel/carboplatin; TbPH: Docetaxel/carboplatin/trastuzumab/pertuzmab; ddAC: dose-dense doxorubicin and cyclophosphamide; * nine patients also treated with trastuzumab; 1 patient treated with bevacizumab.

## Data Availability

Data are available in a publicly accessible repository: https://doi.org/10.6084/m9.figshare.28533173 (accessed on 5 May 2025).

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
