# Peer review of "An Activation Likelihood Estimation Meta-Analysis of Voxel-Based Morphometry Studies of Chemotherapy-Related Brain Volume Changes in Breast Cancer"

_cancers, 2025, doi:10.3390/cancers17101684_

Round 1
Reviewer 1 Report
Comments and Suggestions for Authors
Review on the manuscript of Utecht S et al., (cancers-3541020): “An Activation Likelihood Estimation Meta-Analysis of Voxel Based Morphometry Studies of Chemotherapy Related Brain Volume Changes in Breast Cancer”.
In this study, the Authors conducted a meta-analysis to investigate brain changes in breast cancer patients and survivors who had undergone chemotherapy over time. The Authors show that the right insula and the left inferior frontal gyrus were the regions most consistently observed to have volume reduction over time in breast cancer patients who had undergone chemotherapy. However, other brain regions, including the left medial frontal gyrus, superior frontal gyrus, anterior cingulate, right parahippocampal gyrus, right thalamus, left inferior parietal lobule, right superior parietal lobule, both superior temporal gyrus, and the cerebellum, also showed volumes changes over time in breast cancer patients who had undergone chemotherapy.
Overall, I find this topic to be of great interest, as chemotherapy can have significant effects on cognitive function and brain structure. Thus, understanding structural and functional alterations over time associated with chemotherapy helps identify affected brain regions, and the adoption of interventions to improve cognitive function, emotional well-being, and overall quality of life.
I believe the Authors have addressed the primary question posed. Below, I indicate the issues identified in the current version of the manuscript. I hope the Authors find the following comments and suggestions helpful.
1 - I would kindly suggest that the Authors consider revising the abstract section. In its current form, the information under the background heading does not fully align with what would typically be expected. Moreover, expanding the whole abstract would provide a clearer and more detailed overview of the study.
2 – I suggest that the Authors consider mentioning the different areas highlighted in Figure 3 to enhance the understanding by the readers (the left medial frontal gyrus, superior frontal gyrus, anterior cingulate, right parahippocampal gyrus, right thalamus, left inferior parietal lobule, right superior parietal lobule, both superior temporal gyri, and the cerebellum). The same information would be very informative in Figure 4.
3 - I believe the Authors could consider expanding the discussion section, as it is currently very minimalist and does not explore the results in depth in relation to existing literature. For example, it might be valuable to discuss their findings in comparison with studies on other types of cancer. Additionally, addressing the potential consequences of such reductions in these brain areas could further enrich the discussion.
4 - The Authors conclude the discussion by stating that “Unfortunately, due to the heterogeneity of methodologies across the neuroimaging literature, we were not able to find a large set of homogeneous comparison data to reach the desired study count for a statistically powerful meta-analysis, therefore our meta-analysis results should be considered exploratory”. This suggests that further research is needed in this area. Given the limited data available, it may be worth considering whether a meta-analysis at this stage provides sufficient robustness for publication.
5 - The Authors have structured the Conclusion section on the limitations of the study, which would be more appropriately explored in the Discussion section. In the Conclusion section, it would be beneficial to focus on summarizing the main findings of the study.
6 - I was unable to access the supplementary material referenced in the manuscript.
Minor points:
1 - In Table 2, it would be beneficial if the Authors could include a clarification in the legend regarding the meaning of the information presented in brackets (e.g., SD, SEM?).
Author Response
To my first reviewer,
Thank you for your detailed and organized suggestions. I have addressed each of your points in the list below. Most notably, I have revised the Discussion and Conclusion section and hope the new version better demonstrates how our volumetric findings align with brain function, cognition and the existing literature. I have also created a private link so that you may access the supplementary materials: https://figshare.com/s/2329faff8fc5de1c00ee
1 - I would kindly suggest that the Authors consider revising the abstract section. In its current form, the information under the background heading does not fully align with what would typically be expected. Moreover, expanding the whole abstract would provide a clearer and more detailed overview of the study.
Response: I have revised the abstract to better explain the background of the manuscript.
2 – I suggest that the Authors consider mentioning the different areas highlighted in Figure 3 to enhance the understanding by the readers (the left medial frontal gyrus, superior frontal gyrus, anterior cingulate, right parahippocampal gyrus, right thalamus, left inferior parietal lobule, right superior parietal lobule, both superior temporal gyri, and the cerebellum). The same information would be very informative in Figure 4.
Response: I have added the regions of interest into the text for Figure 4, and moved the text referencing Figure 3.
4 - The Authors conclude the discussion by stating that “Unfortunately, due to the heterogeneity of methodologies across the neuroimaging literature, we were not able to find a large set of homogeneous comparison data to reach the desired study count for a statistically powerful meta-analysis, therefore our meta-analysis results should be considered exploratory”. This suggests that further research is needed in this area. Given the limited data available, it may be worth considering whether a meta-analysis at this stage provides sufficient robustness for publication.
Response: We believe that even with our limited data the consistent findings are worth discussion.
6 - I was unable to access the supplementary material referenced in the manuscript.
I have the public link disabled until publication. Here is a private link that should work. https://figshare.com/s/2329faff8fc5de1c00ee
Minor points:
1 - In Table 2, it would be beneficial if the Authors could include a clarification in the legend regarding the meaning of the information presented in brackets (e.g., SD, SEM?).
Response: I have added this clarification.
3+5:
3 - I believe the Authors could consider expanding the discussion section, as it is currently very minimalist and does not explore the results in depth in relation to existing literature. For example, it might be valuable to discuss their findings in comparison with studies on other types of cancer. Additionally, addressing the potential consequences of such reductions in these brain areas could further enrich the discussion.
5 - The Authors have structured the Conclusion section on the limitations of the study, which would be more appropriately explored in the Discussion section. In the Conclusion section, it would be beneficial to focus on summarizing the main findings of the study.
Response: I have heavily modified the Conclusion and Discussion sections to address these concerns.
I hope the new manuscript effectively satisfies your concerns. Thank you.
Reviewer 2 Report
Comments and Suggestions for Authors
The manuscript by Sonya Utecht and co-authors presents an analysis of the effects of chemotherapy on the brain in breast cancer patients. Chemotherapy is harmful not only to tumour cells but also to normal tissue. There is an effect of this type of anti-cancer therapy on the brain that could be associated with long-term cognitive disorders and neurological symptoms. The author analysed previously published data and drew some conclusions. Although the manuscript may be of interest to some readers of Cancers Journal, there are many things that should be improved before publication. The graphs nicely illustrated the method, but tables with actual numbers in tabular format should be included in the manuscript for comparison. The types of chemotherapy should be accurately described and introduced into the manuscript.
Author Response
To my second reviewer,
Thank you for your response. I know you felt the conclusions needed significant improvement and I hope you find the revised Discussion and Conclusion sections better elaborate the meaning and importance of our findings.
1. The reviewer felt the methods and conclusions sections required improvement.
Response: The conclusion and discussion section have been heavily modified.
2. The graphs nicely illustrated the method, but tables with actual numbers in tabular format should be included in the manuscript for comparison.
Response: The analysis results are graphical showing probability distributions. The peak coordinates used as inputs are in the original papers and the supplementary materials. Here is a private link that to the supplementary materials. https://figshare.com/s/2329faff8fc5de1c00ee
3. The types of chemotherapy should be accurately described and introduced into the manuscript.
Response: I have added chemotherapy information to tables 1 and 2.
Thank you for your review. I hope the revised manuscript satisfies your requests for improvement.
Round 2
Reviewer 1 Report
Comments and Suggestions for Authors
Second review on the manuscript of Utecht S et al., (cancers-3541020): “An Activation Likelihood Estimation Meta-Analysis of Voxel Based Morphometry Studies of Chemotherapy Related Brain Volume Changes in Breast Cancer”.
In this study, the Authors conducted a meta-analysis to investigate brain changes in breast cancer patients and survivors who had undergone chemotherapy over time. The Authors show that the right insula and the left inferior frontal gyrus were the regions most consistently observed to have volume reduction over time in breast cancer patients who had undergone chemotherapy. However, other brain regions, including the left medial frontal gyrus, superior frontal gyrus, anterior cingulate, right parahippocampal gyrus, right thalamus, left inferior parietal lobule, right superior parietal lobule, both superior temporal gyrus, and the cerebellum, also showed volumes changes over time in breast cancer patients who had undergone chemotherapy.
This represents the second version of the manuscript following peer review. The Authors have addressed some of the points raised during the first round of review. However, certain issues still persist and require further attention from the Authors. Below, I outline the issues identified in the current version of the manuscript. I hope the Authors find the following comments and suggestions helpful.
1 - I suggest that the Authors consider point if Figures 3 and 4 the different areas highlighted in Figure 3 to enhance the understanding by the readers (where the left medial frontal gyrus, superior frontal gyrus, anterior cingulate, right parahippocampal gyrus, right thalamus, left inferior parietal lobule, right superior parietal lobule, both superior temporal gyri, and the cerebellum are located).
Minor points:
1 - On line 312, when the Authors mention “inferior partial lobe”, it seems they are referring to the “inferior parietal lobe”. Could the Authors please clarify this point?
Author Response
1 - I suggest that the Authors consider point if Figures 3 and 4 the different areas highlighted in Figure 3 to enhance the understanding by the readers (where the left medial frontal gyrus, superior frontal gyrus, anterior cingulate, right parahippocampal gyrus, right thalamus, left inferior parietal lobule, right superior parietal lobule, both superior temporal gyri, and the cerebellum are located).
Response: We had some trouble understanding but we suspect this is a suggestion to make Figure 3 more similar to Figure 4. However, Figure 3 and Figure 4 are purposefully different in style. Figure 2 is the equivalent to Figure 4 for the BCC+ data, showing the peak consistency in a few select regions. On the other hand Figure 3 allows the viewer to see the expanse of changes over the whole brain. Panel A especially shows several highlighted regions from different angles together to better see the whole effect. This Figure is therefore different in purpose and style than Figure 4.
Minor points:
1 - On line 312, when the Authors mention “inferior partial lobe”, it seems they are referring to the “inferior parietal lobe”. Could the Authors please clarify this point?
This has been corrected.
Reviewer 2 Report
Comments and Suggestions for Authors
The manuscript has been improved, but the supplementary file is not accessible.
Author Response
Thank you for your response.
The manuscript has been improved, but the supplementary file is not accessible.
Response: I am sorry the private link provided did not work for you. I have removed the embargo on the public version of the supplementary data and it should be fully accessible. https://doi.org/10.6084/m9.figshare.28533173
Round 3
Reviewer 1 Report
Comments and Suggestions for Authors
Third review on the manuscript of Utecht S et al., (cancers-3541020): “An Activation Likelihood Estimation Meta-Analysis of Voxel Based Morphometry Studies of Chemotherapy Related Brain Volume Changes in Breast Cancer”.
In this study, the Authors conducted a meta-analysis to investigate brain changes in breast cancer patients and survivors who had undergone chemotherapy over time. The Authors show that the right insula and the left inferior frontal gyrus were the regions most consistently observed to have volume reduction over time in breast cancer patients who had undergone chemotherapy. However, other brain regions, including the left medial frontal gyrus, superior frontal gyrus, anterior cingulate, right parahippocampal gyrus, right thalamus, left inferior parietal lobule, right superior parietal lobule, both superior temporal gyrus, and the cerebellum, also showed volumes changes over time in breast cancer patients who had undergone chemotherapy.
This represents the third version of the manuscript following peer review. The Authors have addressed some of the points raised during the first round of review. However, certain issues still persist and require further attention from the Authors. Below, I outline the issues identified in the current version of the manuscript. I hope the Authors find the following comments and suggestions helpful.
1 - My previous suggestion was for the authors to clearly indicate the different brain regions (left medial frontal gyrus, superior frontal gyrus, anterior cingulate, right parahippocampal gyrus, right thalamus, left inferior parietal lobule, right superior parietal lobule, both superior temporal gyri, and the cerebellum) in Figures 3 and 4. The authors could use arrows and labels to identify each region. Otherwise, readers who are not familiar with the different brain regions may not understand the content of the figures.
Author Response
1 - My previous suggestion was for the authors to clearly indicate the different brain regions (left medial frontal gyrus, superior frontal gyrus, anterior cingulate, right parahippocampal gyrus, right thalamus, left inferior parietal lobule, right superior parietal lobule, both superior temporal gyri, and the cerebellum) in Figures 3 and 4. The authors could use arrows and labels to identify each region. Otherwise, readers who are not familiar with the different brain regions may not understand the content of the figures.
After much consideration we determined the best way to address your issue was to add a neuroanatomical atlas overlay to the figures. Therefore, we have enhanced all the anatomical figures color coded parcellation overlays to clarify the brain regions displayed in each figure. These color overlays make the figures much more informative and visually interesting.
Round 4
Reviewer 1 Report
Comments and Suggestions for Authors
Fourth review on the manuscript of Utecht S et al., (cancers-3541020): “An Activation Likelihood Estimation Meta-Analysis of Voxel Based Morphometry Studies of Chemotherapy Related Brain Volume Changes in Breast Cancer”.
In this study, the Authors conducted a meta-analysis to investigate brain changes in breast cancer patients and survivors who had undergone chemotherapy over time. The Authors show that the right insula and the left inferior frontal gyrus were the regions most consistently observed to have volume reduction over time in breast cancer patients who had undergone chemotherapy. However, other brain regions, including the left medial frontal gyrus, superior frontal gyrus, anterior cingulate, right parahippocampal gyrus, right thalamus, left inferior parietal lobule, right superior parietal lobule, both superior temporal gyrus, and the cerebellum, also showed volumes changes over time in breast cancer patients who had undergone chemotherapy.
This represents the fourth version of the manuscript following peer review. I would like to congratulate the Authors on properly addressing the issues raised during the review rounds.